# Group Cognitive Behavioural Therapy for Non-Rapid Eye Movement Parasomnias: Long-Term Outcomes and Impact of COVID-19 Lockdown

**DOI:** 10.3390/brainsci13020347

**Published:** 2023-02-17

**Authors:** Matthias Laroche, Nazanin Biabani, Panagis Drakatos, Hugh Selsick, Guy Leschziner, Joerg Steier, Allan H. Young, Sofia Eriksson, Alexander Nesbitt, Veena Kumari, Ivana Rosenzweig, David O’Regan

**Affiliations:** 1Sleep and Brain Plasticity Centre, Department of Neuroimaging, Institute of Psychiatry, Psychology and Neuroscience (IoPPN), King’s College London, Box 089, De Crespigny Park, London SE5 8AF, UK; 2Sleep Disorder Centre, Nuffield House, Guy’s Hospital, London SE1 9RT, UK; 3Insomnia and Sleep Medicine Behavioural Clinic, Royal London Hospital for Integrated Medicine, London WCIN 3HR, UK; 4Department of Neurology, Guy’s Hospital, London SE1 9RT, UK; 5Basic and Clinical Neurosciences, Institute of Psychiatry, Psychology and Neuroscience (IoPPN), King’s College London, Box 089, De Crespigny Park, London SE5 8AF, UK; 6Department of Psychological Medicine, Institute of Psychiatry, Psychology and Neuroscience, King’s College London & South London and Maudsley NHS Foundation Trust, Bethlem Royal Hospital, Monks Orchard Road, Beckenham, Kent BR3 3BX, UK; 7National Hospital for Neurology and Neurosurgery, Queen Square, London WC1N 3BG, UK; 8Centre for Cognitive Neuroscience, College of Health, Medicine and Life Sciences, Brunel University London, Uxbridge UB8 3PH, UK; 9Faculty of Life Sciences and Medicine, King’s College, London WC2R 2LS, UK

**Keywords:** cognitive behavioural therapy, CBT, NREM parasomnia, parasomnia, treatment

## Abstract

Prior to the COVID-19 pandemic, we demonstrated the efficacy of a novel Cognitive Behavioural Therapy programme for the treatment of Non-Rapid Eye Movement Parasomnias (CBT-NREMP) in reducing NREM parasomnia events, insomnia and associated mood severities. Given the increased prevalence and worsening of sleep and affective disorders during the pandemic, we examined the sustainability of CBT-NREMP following the U.K.’s longest COVID-19 lockdown (6 January 2021–19 July 2021) by repeating the investigations via a mail survey in the same 46 patient cohort, of which 12 responded. The survey included validated clinical questionnaires relating to NREM parasomnia (Paris Arousal Disorder Severity Scale), insomnia (Insomnia Severity Index) and anxiety and depression (Hospital Anxiety and Depression Scale). Patients also completed a targeted questionnaire (i.e., Impact of COVID-19 Lockdown Questionnaire, ICLQ) to assess the impact of COVID-19 lockdown on NREM parasomnia severity, mental health, general well-being and lifestyle. Clinical measures of NREM parasomnia, insomnia, anxiety and depression remained stable, with no significant changes demonstrated in questionnaire scores by comparison to the previous investigatory period prior to the COVID-19 pandemic: *p* (ISI) = 1.0; *p* (HADS) = 0.816; *p* (PADSS) = 0.194. These findings support the longitudinal effectiveness of CBT-NREMP for up to three years following the clinical intervention, and despite of the COVID-19 pandemic.

## 1. Introduction

To date, there is no homogeneous, standardised model for treating non-rapid eye movement (NREM) parasomnia [1,2,3]. The complexity and phenotypical diversity of underlying neurophysiologic substrates likely contribute to this ambiguity [4]. NREM parasomnia are known as disorders of arousals, or disorders of sleep-state dissociation, with sleepwalking, confusional arousals, night terrors and less well-known entities such as sleep-related sexual behaviours and eating disorders, all relatively prevalent around the globe, and often severe, with significant individual, societal and forensic implications [1,5,6,7,8,9,10].

Traditionally, several fundamental mechanistic concepts, including sleep state instability, sleep inertia or incomplete awakening from NREM sleep, ability to simultaneously exhibit both NREM sleep and wakefulness brain rhythms signatures, and, finally, activation of central pattern motor generators, have been argued to underlie the pathophysiology of the NREM parasomnias (also see [1,4]).

Guided by these mechanistic concepts as useful biomarkers and treatment targets, we have recently developed a novel, group-based, Cognitive Behavioural Therapy (CBT-NREMP) programme [2]. The novel CBT-NREMP programme was specifically designed to incorporate and build-on core principles from the well-established Cognitive Behavioural Therapy for Insomnia (CBT-I) (22). Similar to CBT-I, CBT-NREMP targets co-morbid insomnia, anxiety, stress and other relevant psychological difficulties, all of which have been demonstrated as beneficial in NREMP management (2). Thus, an overarching goal of the CBT-NREMP programme is to target maladaptive sleep-related behaviours, thoughts and anxiety, and therefore, to concomitantly also target those priming and precipitating factors that cause parasomnias to persist over time (for a more in-depth description of the CBT-NREMP programme please refer to [2]).

Moreover, shortly before the start of the COVID-19 pandemic, in a group of NREMP patients, CBT-NREMP was demonstrated as a safe and effective clinical intervention that, due to its unique design, primarily addresses precipitating and perpetuating factors that are behaviourally and psychologically driven [2] (please also refer to Figure 1; CBT-NREMP study).

However, longitudinal robustness and sustainability of CBT-NREMP intervention effects have so far not been investigated. Furthermore, it is unclear whether CBT-NREMP unique design is sufficient to provide longitudinal protection against significant traumatic events, such as the unique and unprecedented bio-psycho-social dimension of the ongoing COVID-19 pandemic [11]. Biological factors such as overt, unknown and latent effects of the infection, vaccines and the impact of long COVID-19 [12] on sleep and mental health [13,14], where relatively little is known about symptom severity, expected clinical course, impact on daily functioning and return to baseline health [15], were expected to have a detrimental impact. Moreover, throughout the pandemic, public health measures, including cessation of international, national and regional travel, lockdowns and restrictions, whilst implemented to reduce community transmission of the virus, have also had recognised neuropsychiatric consequences for the population as a whole, with forced social isolation and personal and professional uncertainty likely leading to an increase in reported insomnia and other sleep disorders, as well as psychiatric symptoms, including stress, anxiety and depression [11,14].

Thus, against this multifactorial aggravating background of the COVID-19 pandemic, we set to monitor, via a mail survey, the sustainability and robustness of the therapeutic effects of CBT-NREMP intervention, approximately three years after the initial treatment. We hypothesised that the temporal stability of the initially demonstrated effects of CBT-NREMP treatment [2] will have been sustained in a group of 46 patients who were treated shortly prior to the start of the pandemic (Figure 1; SURVEY).

For that purpose, the mail survey was undertaken following the U.K.’s longest COVID-19 lockdown (6 January 2021–19 July 2021) (Figure 1; SURVEY), during which the post-pandemic and post-lockdown outcomes on clinical measures of insomnia (Insomnia Severity Index, ISI), clinical severity of NREM parasomnia (Paris Arousal Disorders Severity Scale, PADSS) and anxiety and depression symptoms (Hospital Anxiety and Depression Scale) were collected [2].

## 2. Materials and Methods

### 2.1. Design, Ethics and Data Collection

Forty-six adult patients (*n* = 46) who had completed a complete, five sessions, programme of a structured group CBT-NREMP between November 2018 and January 2020 were contacted via a mail survey, immediately following the longest COVID-19 lockdown period in the U.K. This time point marked approximately three years’ time (Figure 1; SURVEY) after the initial CBT-NREMP treatment (Figure 1; CBT-NREMP study) [2] was delivered. Participants had not engaged in additional talking therapies since completing CBT-NREMP, nor at the time of this study.

Twelve NREM parasomnia patients, 26% of the initial forty-six-patient cohort, completed survey and the questionnaires (Figure 1). Of those, seven patients had a diagnosis of NREM sleep-walking, six suffered with sleep terrors, two with confusional arousals and one patient had a diagnosis of NREM sleep talking, i.e., some patients reported two or more NREM parasomnia subtypes. The patients answered and returned the outcomes of the clinical questionnaires that assess the severity of NREM parasomnia, insomnia and affective symptomatology, i.e., the Insomnia Severity Index (ISI) [16,17], Hospital Anxiety and Depression Scale (HADS) [18] and the Paris Arousal Disorders Severity Scale (PADSS) [5]. The outcomes were collected and analysed, as previously described [2].

All three questionnaires are clinically established self-rated scales, of which PADSS is used to objectively assess the severity of NREM parasomnia symptoms [5]. PADSS has been increasingly used in clinical set ups and in clinical research to assess the efficacy of novel interventions along with potential changes over long periods of time. It is divided in three sections: PADSS-A, which assesses the presence of potentially dangerous parasomnia-related behaviours; PADSS-B, which evaluates the frequency of episodes and PADSS-C, which accounts for the general consequences of the disorder. Similarly, ISI is frequently used as a screening tool to objectively assess the severity and impact of insomnia [16], with scores suggestive of the absence of insomnia (0–7), subthreshold (8–14), moderate (15–21) or severe insomnia (22–28). HADS, another self-rated scale, is commonly used to examine the severity of depression and anxiety symptomology [18] with scores higher than eight in each subscale denoting mild symptoms of the disorder (8–10), a score above eleven suggestive of moderate symptoms (11–14) and a score above fifteen suggesting severe symptoms (15–21).

In order to examine the effect of COVID-19 lockdown restrictions on NREM-parasomnias in a more targeted and precise manner, participants were additionally asked to complete a novel self-report questionnaire, which we developed, i.e., Impact of COVID-19 Lockdown Questionnaire (ICLQ; please see Appendix A). ICLQ has been specifically developed to assess the impact of COVID-19 lockdown restrictions on patients with NREM parasomnia. It consists of four separate subscales, each aiming to target different aspects of sleep, general wellbeing and lifestyle (Appendix A). The first subscale consists of simple ‘Yes’ or ‘No’ questions intended to ascertain the environmental factors contributing to the rater’s lockdown period that invite binary answers; for example, ‘Did you live alone during lockdown’: Yes/No; ‘Did you contract COVID-19′: Yes/No. The second subscale consists of ten Likert ratings scales with answers ranging from one to ten, designed to measure the effect of lockdowns on NREM parasomnia and sleep (1 = Strongly Disagree, 5 = No Change, 10 = Strongly Agree). The possible scores range between ten (Minimal Impact) and 100 (Maximal Impact). The third subscale assesses the effects of lockdown on mental health, focusing on psychological and behavioural aspects of stress, depression and anxiety using the same Likert rating method (1 = Strongly Disagree, 5 = No Change, 10 = Strongly Agree) for which the scores range from 4 (Minimal Impact) to 40 (Maximal Impact). The fourth and final subscale examines the effects of lockdown on general wellbeing and lifestyle, which entails questions regarding physical fitness, substance use and diet, amongst others. The same Likert rating method is used (1 = Strongly Disagree, 5 = No Change, 10 = Strongly Agree), with a possible score ranging between eleven (Minimal Impact) and 110 (Maximal Impact). Although the ICLQ has no previously published psychometric properties, it is Likert-based, with a ‘balanced’ number of options either side of ‘neutral’, with appropriate statistical treatment of these data [19].

### 2.2. Statistical Analysis

Descriptive statistics were used to summarize the data as means ± standard deviation (SD) with median, 25th and 75th percentiles for continuous non-parametric variables. Using the Kolmogorov–Smirnov test of normality, the non-parametric Wilcoxon signed-rank test was conducted to examine the difference in depression, anxiety and each of the sleep-related scores as assessed by self-report questionnaires administered following the CBT-NREMP, as previously described [2], and the scores we collected using the postal survey. The most recent follow-up values were adjusted for multiple tests using Bonferroni correction.

Similarly, in order to investigate the impact of COVID-19 lockdown on NREM parasomnia patients, descriptive statistics were used to summarize the data as means ± standard deviation (SD) with median, 25th and 75th percentiles for continuous variables.

In order to examine the COVID-19 related main effects, one-sample Wilcoxon signed-rank test was conducted to assess whether any significant differences could be found (e.g., significantly different from a score of five which indicates no change in effect).

The analysis of variance (ANOVA) comparisons was also conducted between pre-treatment (before CBT-NREMP intervention), post-treatment (after CBT-NREMP intervention) and the current follow-up (during lockdown) for survey responders. Pairwise comparison of pre-treatment, post-treatment and the current follow-up (during lockdown) time points for the scales of: PADSS, ISI and HADS were conducted using Wilcoxon signed rank test and *p* values were adjusted for multiple tests using Bonferroni correction.

All statistical analyses were conducted using R version 4.1.1.

## 3. Results

### 3.1. Preliminary Findings on Sustainability of the CBT-NREMP Intervention

Altogether twelve patients responded, aged 27 to 58 years-old [mean (SD): 43 (9.8)], 26% of the original cohort [14] (Figure 1), 41.7% male (*n* = 5), and their findings were thus analysed.

Patients were asked to complete follow up ISI, HADS and PADSS and ICL assessments (Figure 1, Appendix A; Table 1 and Table 2). The mean time from entering group CBT-NREMP to follow-up was 962 ± 433 days.

No significant temporal change in the distribution of assessment scores was found by comparison to initial investigations, suggestive of stable effect of the CBT-NREMP intervention, lasting for up to three years following the intervention (Figure 2; *p* = 1 for ISI; *p* = 0.816 for HADS; *p* = 0.194 for PADSS; also see Table 2). Similarly, no significant differences on scores in subscales of HADS and PADSS were demonstrated (Figure 2, *p* = 0.84 for HADS-A, *p* = 1 for HADS-D, *p* = 0.094 for PADSS-A, *p* = 1 for PADSS-B, *p* = 0.719 for PADSS-C).

Outcome values and descriptive statistics are shown in Table 1.

To further analyse our survey responders, ANOVA comparisons were conducted between pre-treatment (before CBT-NREMP intervention), post-treatment (after CBT-NREMP intervention) and the current follow-up (during lockdown). We found no significant difference between pre-treatment, post-treatment and the current follow-up (during lockdown) for scales of: ISI, HADS and PADSS (Table 3 and Appendix A).

### 3.2. Comparisons between Survey Responders and Non-Responders

Given our relatively low number of survey responders (*n* = 12), we sought to compare them to the non-responders (*n* = 34) with regard to social demographics, as well as baseline: NREM parasomnia, insomnia, anxiety and depression severities. Responders and non-responders were comparable across gender proportions, as well as baseline PADSS, ISI and HADS severities (Table 4). Responders tended to be slightly older, which reached statistical significance (*p* = 0.042).

We further sought to compare non-responders’ results pre-treatment (before CBT-NREMP intervention) and post-treatment (after CBT-NREMP intervention) for scales of: ISI, HADS and PADSS. For non-responders, there were significant differences in scores of ISI (*p* = 0.037) and PADSS (*p* = 0.016) scales between pre-treatment (before CBT-NREMP intervention) and post-treatment (after CBT-NREMP intervention) time points (Table 5).

### 3.3. Impact of COVID-19 Lockdowns as Explored by ICLQ

Patients were additionally asked to complete the ICLQ (please refer to Appendix A), designed to retrospectively assess the effects of COVID-19 imposed lockdown on NREM parasomnia symptoms, mental health, general wellbeing and lifestyle (Table 6 and Appendix A; Appendix A).

Outcome values and descriptive statistics are shown in Table 6.

## 4. Discussion

These findings demonstrate a remarkable robustness and longitudinal sustainability of up to three years of the beneficial effects of the CBT-NREMP intervention in a cohort of NREM parasomnia patients (Figure 2 and Appendix A, Table 2). We previously demonstrated that five weeks of a structured group CBT intervention in adult patients with NREM parasomnia can lead to a significant reduction in its severity [2], and here we demonstrate that this can be sustained up to three years, even in the backdrop of significant precipitating and perpetuating factors exerted during the course of the COVID-19 pandemic. In our study, this is evident by a sustained reduction in total PADSS and PADSS-A patients’ scores (Table 2), known to closely correlate with video-polysomnography-ascertained severity of parasomnia behaviours [2,5]. Similar sustained beneficial effects were demonstrated for the patients’ insomnia and affective symptomatology, as evidenced by the ISI and HADS scores (Table 2).

Notably, our findings, as evidenced by the ICLQ scores (Table 6 and Appendix A; Appendix A), do suggest a negative impact of the UK’s longest COVID-19 lockdown on patients’ sleep and well-being. This effect appears to have been predominantly driven by (subjective) deteriorations in sleep quality and quantity, as well as down to raised anxiety and affective baseline (Appendix A). This is in keeping with previous studies where negative psychosocial changes have been shown to have a recurring negative impact on sleep, mental health and lifestyle habits, all of which can adversely affect parasomnia phenotypes [2,20,21]. For example, one study which relied on survey data from over 45,000 U.K. adults, found that factors stemming from lockdown measures such as illness due to COVID-19, financial difficulty, loss of paid work, difficulties acquiring medication, accessing food and threats to personal safety, all contributed to poorer sleep quality and mental health deterioration [14]. Moreover, as we have seen with other pandemics, sleep and daytime dysfunction persist long after the threat of infection has passed [11,22].

Even so, our patient cohort remained overall clinically stable (Table 1; Figure 2). Arguably, this may be due to ongoing effects of CBT-NREMP intervention, which, much as other CBT programmes for sleep disorders are [23], is based on the premise of addressing behavioural and psychological components that precipitate and perpetuate the disorder [2]. Therefore, at times of adversity, CBT-NREMP tools can be (and presumably have been) transitioned into self-administered habits that target NREM parasomnia behavioural and psychological priming and precipitating factors [2,18]. Effectively, CBT-NREMP intervention assigns an active role to patients, providing the basis for a more independent treatment framework. This notion resonates with the concept of patient activation and empowerment [24], in which treatment modalities bolster patients’ willingness and ability to adopt a more proactive approach in response to their health issues and adherence to treatments. This finding adds to the body of work which suggests that CBT models for treating sleep disorders may also promote sleep and health resilience [25,26]. On the other hand, by and large, such an approach is limited during pharmacotherapeutic interventions for NREM parasomnia.

We acknowledge that the size of our survey responder cohort (*n* = 12; i.e., 26% of the original cohort) is a limitation. However, it should be noted that a 26% mail survey response rate is actually more than double the published expected rates of reply [26]. Our survey responders also tended to be older (which reached statistical significance; Table 4), and hence may have been more likely to respond to a mail survey [27]. Moreover, although survey responders had similar baseline NREM parasomnia, insomnia and affective baseline severities as survey non-responders (Table 4), they did not fare as well post-CBT-NREMP (Table 3, Table 5 and Appendix A). It is therefore tempting to speculate that the survey responders were also more refractory to treatment, and hence more likely to respond to the mail survey [28,29].

## 5. Conclusions

In summary, despite its limitation, including the size and method of a delivery that precluded any significant generalisations and assessments of causality, our study supports the CBT-NREMP intervention as an economical, sustainable and generalisable treatment modality. All the self-reported scores recorded in the PADSS, ISI and HADS have long been robustly validated for clinical use [5,16,18] with the primary goal to enable treatment and monitoring of patients’ symptoms per their own criteria [3]. Finally, we acknowledge that there may have been a responder bias, i.e., that those who were older, and who fared less well with CBT-NREMP were more inclined to respond.

Future research should implement larger randomised controlled trials in order to reliably confirm and expand on the current findings. Such research would ideally include multi-centre neuroimaging and physiological investigations in order to understand the mechanisms by which CBT-NREMP intervention reduces parasomnia severity.

## Figures and Tables

**Figure 1 brainsci-13-00347-f001:**
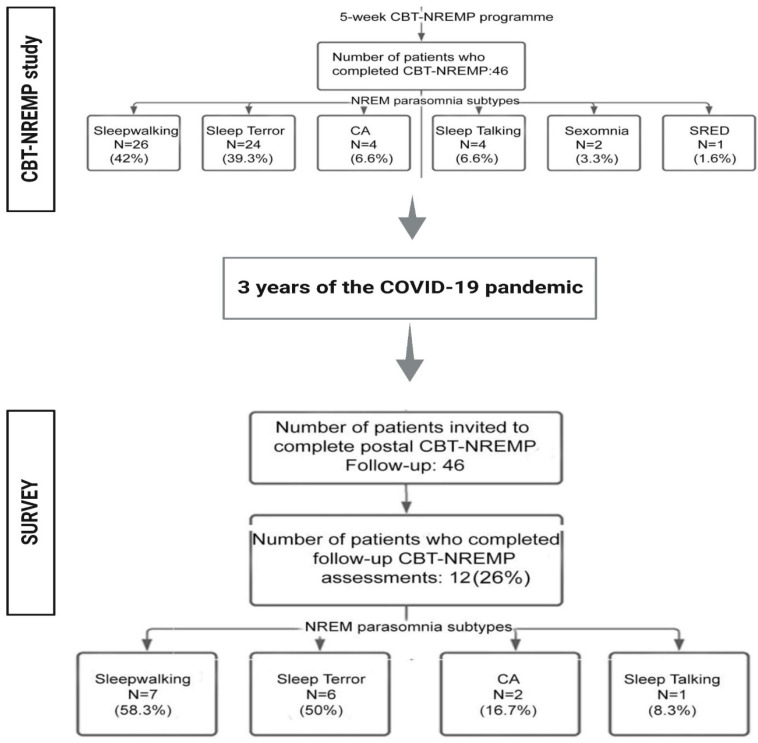
Flowchart of the studied cohort of patients with NREM parasomnia. Some patients reported two or more subtypes of NREM parasomnia. Percentages indicate the prevalence of each NREM parasomnia subtype in our cohort. CBT-NREMP: Cognitive Behavioural Therapy for NREM parasomnia; CA: confusional arousal; SRED: sleep-related eating disorder; NREM: Non-REM.

**Figure 2 brainsci-13-00347-f002:**
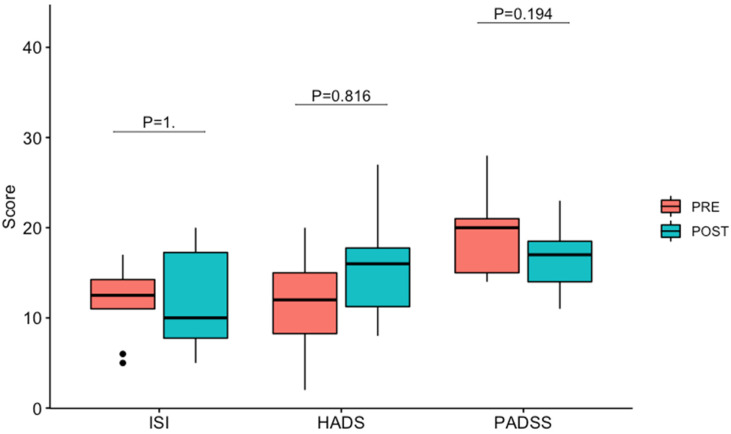
Box plot of Assessment Score for PRE and POST scores for ISI, HADS and PADSS (note: *p* values are from Wilcoxon tests). PRE: after CBT-NREMP intervention; POST: follow-up. Box represents the 50% of the central data (between 25th and 75th percentiles), with a line inside that represents the median. Dots represent points outside 1.5 times the interquartile range above the upper quartile and below the lower quartile.

**Table 1 brainsci-13-00347-t001:** Outcomes of ISI, HADS and PADSS assessment after CBT-NREMP intervention (PRE) and subsequent postal survey findings (POST).

Assessment *n* = 12	PRE		POST	
	Mean (SD)	Median (Q1, Q3)	Mean (SD)	Median (Q1, Q3)
ISI	12.17 (3.71)	12.5 (11, 14.25)	11.75 (5.5)	10 (7.75, 17.25)
HADS	11.25 (5.15)	12 (8.25, 15)	15.9 (6.01)	16 (11.25, 17.75)
HADS_A	5 (2.52)	5.5 (3, 6)	6.9 (2.18)	8 (6, 8)
HADS_D	6.25 (4.18)	6.5 (3, 9.25)	9 (4.52)	8 (6.25, 10)
PADSS	19.08 (4.25)	20 (15, 21)	16.67 (3.58)	17 (14, 18.5)
PADSS_A	9.42 (3.85)	9.5 (6.5, 11.25)	7.08 (3.23)	8 (4.5, 9.25)
PADSS_B	4.42 (1)	4 (4, 5)	4 (1.54)	4 (4, 4.25)
PADSS_C	5.25 (1.82)	5 (4, 7)	4.27 (2)	5 (3.5, 5.5)

Abbreviations: HADS, Hospital Anxiety and Depression Scale (total score); HADS-A, Hospital Anxiety and Depression Scale-Anxiety subset score; HAD-D, Hospital Anxiety and Depression Scale—Depression subset score; ISI, Insomnia Severity Index; PADSS, Paris Arousal Disorders Severity Scale (total score); PADSS-A, Paris Arousal Disorders Severity Scale-subset A score; PADSS-B; Paris Arousal Disorders Severity Scale subset-B score; PADSS-C, Paris Arousal Disorders Severity Scale subset-C score. Q1, 25% percentile. Q3, 75% percentile. SD, standard deviation.

**Table 2 brainsci-13-00347-t002:** Results of Wilcoxon signed-rank tests comparing two time points (*PRE*, initial test scores following the group CBT-NREMP intervention and *POST*, subsequent postal survey findings) and scores for ISI, HADS and PADSS assessments.

PRE	POST	Different from PRE to POST Median (Q1, Q3)	Wilcoxon Signed-Rank Test **	*p* *
SI.Pre	ISI.Post	−2 (−5.25, 4.5,)	42	1
HADS.Pre	HADS.Post	3 (−2, 6.25)	11	0.816
HADS_A.Pre	HADS_A.Post	2 (0, 3.5)	6	0.84
HADS_D.Pre	HADS_D.Post	0.5 (−1.5, 5.5)	12	1
PADSS.Pre	PADSS.Post	−2.5 (−4, −0.75)	50	0.194
PADSS_A.Pre	PADSS_A.Post	−2 (−4, −0.75)	52.5	0.094
PADSS_B.Pre	PADSS_B.Post	0 (−0.25, 0)	6	1
PADSS_C.Pre	PADSS_C.Post	0 (−1.5, 0)	19	0.719

Abbreviations: ISI, Insomnia Severity Index; PADSS, Paris Arousal Disorders Severity Scale (total score); PADSS-A, Paris Arousal Disorders Severity Scale-subset A score; PADSS-B; Paris Arousal Disorders Severity Scale subset-B score; PADSS-C, Paris Arousal Disorders Severity Scale subset-C score; HADS, Hospital Anxiety and Depression Scale (total score); HADS-A, Hospital Anxiety and Depression Scale-Anxiety subset score; HAD-D, Hospital Anxiety and Depression Scale—Depression subset score. Statistically significant values are shown. * Adjusted with Bonferroni correction. ** Wilcoxon signed-rank test was conducted on non-missing observations having both PRE and POST data points.

**Table 3 brainsci-13-00347-t003:** Results of ANOVA test between pre-treatment (before CBT-NREMP intervention), post-treatment (after CBT-NREMP intervention) and the current follow-up (during lockdown) for scales of ISI, HADS and PADSS for responders (*n* = 12).

	Responders (*n* = 12)	
	F Statistics (Degree of Freedom)	*p*-Value
ISI	0.495	0.614
HADS	1.654	0.208
HADS_A	1.495	0.240
HADS_D	0.946	0.399
PADSS	1.938	0.160
PADSS_A	2.627	0.087
PADSS_B	0.500	0.611
PADSS_C	0.941	0.401

Abbreviations: ISI, Insomnia Severity Index; PADSS, Paris Arousal Disorders Severity Scale (total score); PADSS-A, Paris Arousal Disorders Severity Scale-subset A score; PADSS-B; Paris Arousal Disorders Severity Scale subset-B score; PADSS-C, Paris Arousal Disorders Severity Scale subset-C score; HADS, Hospital Anxiety and Depression Scale (total score); HADS-A, Hospital Anxiety and Depression Scale-Anxiety subset score; HAD-D, Hospital Anxiety and Depression Scale—Depression subset score.

**Table 4 brainsci-13-00347-t004:** Comparison of demographics and scales of ISI, HADS and PADSS between responders and non-responders.

	Non-Responders (*n* = 34)	Responders (*n* = 12)	*p*-Value
Gender (%)			
Male	19 (55.9)	7 (58.3)	*p* = 1.000 *
Female	15 (44.1)	5 (41.7)	
Age—mean (SD)	32.8 (10.1)	39.8 (9.4)	*p* = 0.042 **
Pre—mean (SD)			
ISI	15.2 (4.5)	13.5 (4.1)	*p* = 0.251 **
HADS	14.5 (6.2)	12.9 (6.8)	*p* = 0.480 **
HADS_A	6.7 (4.7)	5.8 (2.9)	*p* = 0.449 **
HADS_D	7.9 (4.4)	7.2 (5.3)	*p* = 0.686 **
PADSS	19.2 (6.4)	20.1 (5.1)	*p* = 0.636 **
PADSS_A	9.4 (4.7)	10.6 (4.3)	*p* = 0.436 **
PADSS_B	4.5 (1.2)	4.4 (0.9)	*p* = 0.877 **
PADSS_C	5.3 (1.8)	5.1 (1.6)	*p* = 0.690 **
Post—mean (SD)			
ISI	12.8 (4.2)	12.2 (3.7)	*p* = 0.658 **
HADS	13.3 (6.5	11.2 (5.2)	*p* = 0.284 **
HADS_A	6.3 (4.4)	5 (2.5)	*p* = 0.238 **
HADS_D	7 (3.8)	6.2 (4.2)	*p* = 0.581 **
PADSS	17 (5.6)	19.1 (4.3)	*p* = 0.203 **
PADSS_A	8 (4.2)	9.4 (3.8)	*p* = 0.304 **
PADSS_B	4.3 (1.2)	4.4 (1)	*p* = 0.819 **
PADSS_C	4.6 (2)	5.2 (1.8)	*p* = 0.344 **

Abbreviations: ISI, Insomnia Severity Index; PADSS, Paris Arousal Disorders Severity Scale (total score); PADSS-A, Paris Arousal Disorders Severity Scale-subset A score; PADSS-B; Paris Arousal Disorders Severity Scale subset-B score; PADSS-C, Paris Arousal Disorders Severity Scale subset-C score; HADS, Hospital Anxiety and Depression Scale (total score); HADS-A, Hospital Anxiety and Depression Scale-Anxiety subset score; HAD-D, Hospital Anxiety and Depression Scale—Depression subset score. Statistically significant values are shown. * Pearson’s Chi-squared test with Yates’ continuity correction. ** Student’s *t*-test.

**Table 5 brainsci-13-00347-t005:** Comparisons of pre-treatment (before CBT-NREMP intervention) and post-treatment (after CBT-NREMP intervention) timepoints for scales of: ISI, HADS and PADSS, using Wilcoxon signed rank test for non-responders (*n* = 34).

Scale	Wilcoxon Signed-Rank Test Statistic	*p* *
ISI	249.5	0.037 *
HADS	184	0.509
HADS_A	157	1
HADS_D	234	0.434
PADSS	203.5	0.016 *
PADSS_A	248.5	0.159
PADSS_B	46	1
PADSS_C	146.5	0.289

Abbreviations: ISI, Insomnia Severity Index; PADSS, Paris Arousal Disorders Severity Scale (total score); PADSS-A, Paris Arousal Disorders Severity Scale-subset A score; PADSS-B; Paris Arousal Disorders Severity Scale subset-B score; PADSS-C, Paris Arousal Disorders Severity Scale subset-C score; HADS, Hospital Anxiety and Depression Scale (total score); HADS-A, Hospital Anxiety and Depression Scale—Anxiety subset score; HAD-D, Hospital Anxiety and Depression Scale—Depression subset score. Statistically significant values are shown. * Wilcoxon signed rank test along with adjusted *p* values for multiple tests using Bonferroni correction.

**Table 6 brainsci-13-00347-t006:** Descriptive statistics for ICLQ variables. Descriptive statistics (*n* = 12) on depression, anxiety and each of the sleep-related scores during COVID-19 Lockdown using one sample Wilcoxon signed-rank test. A score of above 5 indicates change in the effect.

Variable	n (Not Missing)	Mean (SD)	Median (Q1, Q3)
NREM events deteriorated	12	6.17 (1.34)	6.5 (5, 7)
Sleep deteriorated	12	6.08 (1.78)	6 (5, 7.25)
Lockdown continues to adversely affect my sleep	12	5.08 (2.11)	5 (3.75, 6.25)
Increased sleep onset latency	12	5 (2.37)	5 (4.5, 7)
Increased wake-after-sleep-onset	12	5 (2.52)	5 (3, 7)
Difficult to rise on time	12	7.25 (2.18)	7 (5, 9.25)
Excessive daytime somnolence	12	4.92 (2.61)	5 (3, 6.25)
Poor sleep quality	12	6.25 (2.6)	7 (4.75, 8)
Increased daytime tiredness	12	6.5 (2.28)	7 (5, 8)
Overactive mind at night	12	7.08 (1.78)	7.5 (5.75, 8)
Mood deteriorated	12	6.25 (2.42)	7 (4.75, 8)
Increased anxiety	12	6.92 (2.35)	7 (6, 8.25)
Increased depression	12	6.17 (2.52)	7 (5.75, 7.25)
Increased stress	12	7.17 (2.62)	8 (7, 8.25)
Reduced engagement with exercise	12	5.75 (3.02)	7 (2.75, 8)
Increased alcohol consumption	12	6.17 (2.21)	6.5 (5, 8)
Increased illicit substance use	12	3.25 (2.01)	4.5 (1, 5)
Poor diet	12	6.42 (1.68)	7 (6, 7.25)
Increased loneliness	12	5.5 (2.24)	5.5 (5, 7.25)
Relationships were adversely affected	12	6.17 (2.72)	7 (5.75, 8)
Ability to communicate with others were adversely affected	12	6.17 (2.72)	7 (5.75, 8)
Finances were adversely affected	12	5 (1.86)	5 (4, 6)
Working from home had a negative impact on my general wellbeing	12	6.08 (2.35)	6 (5.75, 7.25)
My responsibilities increased	12	5 (2.86)	5 (4, 7)
The negative effects of lockdown continue to impact my general wellbeing	12	5.67 (2.23)	6 (5, 7.25)

Abbreviations: ICLQ = Impact of COVID-19 Lockdown Questionnaire.

## Data Availability

Original data may be obtained by contacting the corresponding author.

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
