# Peer review of "Group Cognitive Behavioural Therapy for Non-Rapid Eye Movement Parasomnias: Long-Term Outcomes and Impact of COVID-19 Lockdown"

_brainsci, 2023, doi:10.3390/brainsci13020347_

Round 1
Reviewer 1 Report
This study assesses the stability of the effects of a novel Cognitive Behavioural Therapy programme for the treatment of Non-Rapid Eye Movement Parasomnias (CBT- NREMP). Outcome variables were self-reported test measuring NREM parasomnia events, insomnia and associated mood severities. This study was carried out during the COVID-19 lockdown by using a mail survey
26% of the original sample participated in the study and showed no significant changes if clinical measures of NREM parasomnia, insomnia, anxiety and depression.
Thie study in undoubetdly interetsing, although there some major limitations, which have been mostly mentioned by the authors.
Although their basic limitations, some points will make this study publishable:
1. data and statistical comparisons should also include the pre-treatment measures; hence, the results have to report ANOVA comparisons between pre-treatment, post-treatment and the current follow-up
2. due to the extent of the potential responder bias (26% of the original sample participated to this follow-up), comparison between responders and non-responders on social demographics, NREM parasomnia symptom severity, insomnia, anxiety and depression have to be reported. These comparisons have to be reported in relation to both pre-treatment and post-treatments phases
3. data on ICLQ variables have no real significance or relevance, and the one sample Wilcoxon signed-rank test yields very few information. This part has to be roemoved or, at least, shortly reported as descriptive statistics
Author Response
Dear Reviewer,
Thank you very much for your helpful review of our manuscript - Group Cognitive Behavioural Therapy for Non-Rapid Eye Movement Parasomnias: Long-term Outcomes and Impact of COVID-19 Lockdown. We appreciate your valuable comments and would like to thank you for the effort and time you spent improving our manuscript. We have carefully responded to each comment on a point-by-point basis. Please find our responses below, and appropriate in-text modifications (outlined in red).
Yours sincerely,
David O’Regan, on behalf of the authors.
- data and statistical comparisons should also include the pre-treatment measures; hence, the results have to report ANOVA comparisons between pre-treatment, post-treatment and the current follow-up.
- Many thanks for this excellent comment. The additional ANOVA analysis has been completed, and is included in the body of the paper, highlighted in red for ease.
- due to the extent of the potential responder bias (26% of the original sample participated to this follow-up), comparison between responders and non-responders on social demographics, NREM parasomnia symptom severity, insomnia, anxiety and depression have to be reported. These comparisons have to be reported in relation to both pre-treatment and post-treatments phases.
- This has now been completed, and in included in the paper and supplement, highlighted in red for ease.
- data on ICLQ variables have no real significance or relevance, and the one sample Wilcoxon signed-rank test yields very few information. This part has to be removed or, at least, shortly reported as descriptive statistics
- We have made these recommended changes, included in the paper and supplement, and highlighted in red for ease.

Reviewer 2 Report
Manuscript ID: brainsci-2130329
Title: Group Cognitive Behavioural Therapy for Non-Rapid Eye Movement Parasomnias: Long-term Outcomes and Impact of COVID-19 Lockdown.
Journal: Brain Sciences
General comments
1. My major concerns refer to the extremely limited sample size, i.e., n=12, and the very low participation rate, i.e., 26%.
2. In line with my previous concerns, there is a risk that the statistical analyses may be affected by the extremely limited sample size. Moreover, considering the participation rate, the representativity of the sample is somewhat questionable.
Specific comments
Abstract
1. Line 29. Authors should also report the number of participants that answered to the mail survey and were therefore included in the study, i.e., n=12.
Introduction
1. Lines 57-58. I am wondering whether, meanwhile, the CBT-NREMP has been used also by other research/clinical groups. If so, the corresponding studies should be quoted.
Material and Methods
1. Lines 110-112. Authors should clarify that few patients received more than one diagnosis. This is the reason why the sum of number there reported is not 12. This also applies to Figure 1.
2. Lines 134-135. Authors should provide more information about the ICLQ. Who did develop this measure? Are there any published data about its psychometric properties?
Results
1. Table 1. I would suggest that Authors also consider the scores before the beginning of the treatment, comparing data collected at three time points.
2. Line 189. Authors wrote “FU, follow-up” but it seems to me that this acronym does not appear within the table.
Discussion
1. Lines 266-268. Authors wrote “we demonstrate that five weeks of a structured group CBT intervention in adult patients with NREM parasomnia can lead to a significant reduction in its severity”. However, it seems to me that the data supporting this statement are not shown.
Conclusions
1. Lines 316-317. Authors wrote that “there were no differences in terms of social demographics nor NREM parasomnia symptom severity between responders and non-responders.”. I would suggest that Authors show the data supporting this statement.
Author Response
Dear Reviewer,
Thank you very much for your helpful review of our manuscript - Group Cognitive Behavioural Therapy for Non-Rapid Eye Movement Parasomnias: Long-term Outcomes and Impact of COVID-19 Lockdown. We appreciate your valuable comments and would like to thank you for the effort and time you spent improving our manuscript. We have carefully responded to each comment on a point-by-point basis. Please find our responses below, and appropriate in-text modifications (outlined in red).
Yours sincerely,
David O’Regan, on behalf of the authors.
- My major concerns refer to the extremely limited sample size, i.e., n=12, and the very low participation rate, i.e., 26%.
- Many thanks for your comment, and we have addressed this to the best of our ability, undertaking additional analyses between survey responders and non-responders, as highlighted in the paper and and supplement (outlined in red for ease).
- In line with my previous concerns, there is a risk that the statistical analyses may be affected by the extremely limited sample size. Moreover, considering the participation rate, the representativity of the sample is somewhat questionable.
- Many thanks for this excellent comment, and we have undertaken additional analyses, and further compared survey responders and non-responders at each time point i.e. pre-CBT-NREMP, post-CBT-NREMP, and at this follow-up point (see paper and supplement; additional analyses outlined in red). We hope that this adequately addresses this concern, which we have further highlighted as a limitation of our manuscript.

Reviewer 3 Report
Group Cognitive Behavioural Therapy for Non-Rapid Eye 2 Movement Parasomnias: Long-term Outcomes and Impact of 3 COVID-19 Lockdown is an interesting paper, but
1. the sample size of the study is too small
2. the results are too short - the have to extented (tab. & fig. aren't enough)
3 the discussion has to extended too.
Author Response
Dear Reviewer,
Thank you very much for your helpful review of our manuscript - Group Cognitive Behavioural Therapy for Non-Rapid Eye Movement Parasomnias: Long-term Outcomes and Impact of COVID-19 Lockdown. We appreciate your valuable comments and would like to thank you for the effort and time you spent improving our manuscript. We have carefully responded to each comment on a point-by-point basis. Please find our responses below, and appropriate in-text modifications (outlined in red).
Yours sincerely,
David O’Regan, on behalf of the authors.
- the sample size of the study is too small
- We have undertaken additional analyses between survey responders and non-responders (please see manuscript and supplement; changes highlighted in red). We have also emphasised this as a limitation of the study, and made recommendations for further studies, where this can be additionally addressed.
- the results are too short - the have to extented (tab. & fig. aren't enough)
- We have extended the results section (see manuscript and supplement) to address this issue.
- the discussion has to extended too.
- We have also extended our discussion, as highlighted in red in the manuscript.

Round 2
Reviewer 1 Report
The authors responded and made revisions according to the points I raised. It seems now publishable
Reviewer 3 Report
“Group Cognitive Behavioural Therapy for Non-Rapid Eye Movement Parasomnias: Long-term Outcomes and Impact of COVID-19 Lockdown” is edited in sense of reviewers and can be pulished after minor corrections.